# Aerobic Exercise Training and In Vivo Akt Activation Counteract Cancer Cachexia by Inducing a Hypertrophic Profile through eIF-2α Modulation

**DOI:** 10.3390/cancers14010028

**Published:** 2021-12-22

**Authors:** Marcelo G. Pereira, Vanessa A. Voltarelli, Gabriel C. Tobias, Lara de Souza, Gabriela S. Borges, Ailma O. Paixão, Ney R. de Almeida, Thomas Scott Bowen, Marilene Demasi, Elen H. Miyabara, Patricia C. Brum

**Affiliations:** 1School of Physical Education and Sport, University of Sao Paulo, Sao Paulo 05508030, Brazil; vavoltarelli@gmail.com (V.A.V.); gabrielcardialtobias@hotmail.com (G.C.T.); lardesouza@usp.br (L.d.S.); gabrielaborges@usp.br (G.S.B.); ailmaoliveira@usp.br (A.O.P.); neyalmei@usp.br (N.R.d.A.); 2Leeds School of Biomedical Sciences, Faculty of Biological Sciences, University of Leeds, Leeds LS2 9JT, UK; T.S.Bowen@leeds.ac.uk; 3Sirio-Libanes Hospital, Sao Paulo 01308050, Brazil; 4Department of Surgery, Beth Israel Deaconess Medical Center, Harvard Medical School, Boston, MA 02115, USA; 5Children’s Cancer and Blood Foundation Laboratories, Departments of Pediatrics, and Cell and Developmental Biology, Drukier Institute for Children’s Health, Meyer Cancer Center, Weill Cornell Medicine, New York, NY 10021, USA; 6Biochemistry and Biophysics Laboratory, Butantan Institute, Sao Paulo 05503900, Brazil; marilene.demasi@butantan.gov.br; 7Department of Anatomy, Institute of Biomedical Sciences, University of Sao Paulo, Sao Paulo 05508000, Brazil; elenm@usp.br

**Keywords:** cancer cachexia, skeletal muscle plasticity, physical exercise, Akt/mTORC1 signaling, Akt-induced hypertrophy, eIF-2α, translation initiation

## Abstract

**Simple Summary:**

Chronic disease-related muscle atrophy is a serious public health problem since it reduces mobility and contributes to increases in hospitalization costs. Unfortunately, there is no approved treatment for muscle wasting at present. Thus, an understanding of the mechanisms underlying the control of muscle mass and function under chronic diseases can pave the way for the discovery of innovative therapeutic strategies to counteract muscle wasting. Since numerous types of cancer induce cachexia, which has no cure nor an effective treatment, the main proposal here was to study the effects of AET in cancer cachexia, and to investigate, through in vivo manipulation of the Akt/mTORC1 pathway, whether the cachectic muscle still presents conditions to respond adaptively to hypertrophic stimuli. Our results could provide a basis for innovative research lines to better understand muscle plasticity and to investigate potential therapeutic approaches necessary to prevent muscle loss.

**Abstract:**

Cancer cachexia is a multifactorial and devastating syndrome characterized by severe skeletal muscle mass loss and dysfunction. As cachexia still has neither a cure nor an effective treatment, better understanding of skeletal muscle plasticity in the context of cancer is of great importance. Although aerobic exercise training (AET) has been shown as an important complementary therapy for chronic diseases and associated comorbidities, the impact of AET on skeletal muscle mass maintenance during cancer progression has not been well documented yet. Here, we show that previous AET induced a protective mechanism against tumor-induced muscle wasting by modulating the Akt/mTORC1 signaling and eukaryotic initiation factors, specifically eIF2-α. Thereafter, it was determined whether the in vivo Akt activation would induce a hypertrophic profile in cachectic muscles. As observed for the first time, Akt-induced hypertrophy was able and sufficient to either prevent or revert cancer cachexia by modulating both Akt/mTORC1 pathway and the eIF-2α activation, and induced a better muscle functionality. These findings provide evidence that skeletal muscle tissue still preserves hypertrophic potential to be stimulated by either AET or gene therapy to counteract cancer cachexia.

## 1. Introduction

Cachexia is a consequence of many chronic diseases, such as cancer [1], heart failure [2], diabetes [3], and many incurable myopathies affecting children [4]. Cancer-related cachexia affects around 80% of patients and accounts for up to 30% of deaths [5]. These significant numbers make cachexia one of the main paraneoplastic syndromes affecting cancer patients’ quality of life, in addition to reducing tolerance and responsiveness to treatment [6]. Despite continuous improvements in cancer prognosis, cachexia, as well as all other forms of pathology-related muscle atrophy, still have neither a cure nor an effective treatment.

In this context, a combination of interventions that emphasize the importance of regular exercise training has been proposed as a complementary therapy, since it is widely recognized that aerobic exercise training (AET) can act in a preventive and/or therapeutic way in several non-communicable chronic diseases [7,8,9]. It has been shown that AET reduces tumor growth, thus increasing survival in experimental models of cancer [10,11]. More recently, our group showed that AET counteracts cancer cachexia-induced oxidative stress [12]. Indeed, further muscle proteomic analyses have revealed eukaryotic initiation factor 2 (eIF2) signaling and ribosomal proteins to be significantly affected in tumor-bearing animals, which brings forward the hypothesis that cancer-related muscle wasting is not only associated with increases in proteolysis, but also with an impaired anabolic capacity by a dysregulation in the translation initiation process.

Better understanding of the signaling pathways that control muscle mass and function in the context of cancer cachexia is of great scientific relevance. The major signaling pathway regulating adult muscle mass and function is the Akt/mTORC1 pathway [13,14], which is thought to act mainly through increases in protein synthesis by modulating translation initiation [15]. In fact, we have recently reported, by comparison among different models of muscle growth, that Akt/mTORC1 pathway activation and translation initiation are the key processes of hypertrophy [16]. Due to its complexity, this signaling pathway can be modulated by several ways, suggesting that there are potential intrinsic targets for therapeutic approaches and drug development to counteract muscle wasting [17]. In this sense, it was previously shown that the in vivo activation of Akt is able to revert muscle atrophy induced by denervation [18]. However, it is worth highlighting that Akt-induced hypertrophy has never been tested in cancer cachexia.

Taking into consideration that we have previously observed that cancer cachexia induces a dysfunction in the translation initiation process that might be counteracted by AET, our hypotheses in the present study were twofold. We first hypothesized that AET would prevent cancer cachexia by modulating both the Akt/mTORC1 signaling pathway and eIF2 functionality. Secondly, we hypothesized that the in vivo activation of Akt would revert cancer-related muscle wasting. Our main findings indicate that AET, when applied previously to cancer development, prevented cancer cachexia by modulating eIF-2α protein expression. Moreover, Akt-induced hypertrophy also reverted cancer-related muscle wasting by regulating the eIF-2α protein expression, in addition to inducing better muscle functionality.

## 2. Materials and Methods

This study was carried out in accordance with the ethical principles for animal research set forth by the Brazilian National Council for Animal Experiment Control. All protocols were approved by the Ethics in Animal Research Committee of the School of Physical Education and Sports at the University of Sao Paulo (Permit Number: 2017/01).

### 2.1. Animal Model

Adult Balb/C mice (12- to 16-week-old) were used in the present study. Mice were housed in standard plastic cages and kept in an animal facility in a temperature- and light-controlled environment (23 °C; 12/12-h light/dark cycle), with *ad libitum* access to standard mouse chow and water. The sample size used in each experimental procedure is indicated in each figure legend.

To induce cancer cachexia, CT26 colon adenocarcinoma cells were used [19]. CT26 tumor cells were cultured in RPMI 1640 ATCC modification medium (Gibco, ThermoFischer Scientific, Waltham, MA, USA) supplemented with 10% fetal bovine serum (Gibco) and 1% penicillin-streptomycin at 37 °C with 5% CO_2_. After reaching around 80% of confluence, cells were harvested using trypsin (0.05%, Gibco) and centrifugated at 1000× *g* for 5 min. Viable CT26 cells (1.0 × 10^6^) diluted in 100 μL of serum-free RPMI medium were subcutaneously injected into the mice’s right flank as previously described [12]. The mice from our control group were just injected with the same volume of serum-free media. To estimate the tumor volume, the largest and smallest tumor diameters were measured and both values obtained were used in a calculation as previously described [20]. At the end of each experimental procedure, mice were euthanized by cervical dislocation under isoflurane anesthesia (Isoforine^®^, Cristalia, Brazil). Tissue samples were immediately harvested and weighed, then stored at −80 °C for further analysis.

### 2.2. Physical Exercise

#### 2.2.1. Maximal Running Capacity Test (MRC)

MRC test was performed on a motor treadmill (AVS Projetos, São Carlos, Brazil) using a graded exercise protocol previously established by our group [21]. Briefly, after 5 consecutive days (10 min per day) of adaptation period, mice were placed in the treadmill, and the exercise intensity, starting at 6 m/min, was increased by 3 m/min every 3 min at 0% grade until mice were able to run no longer. Two MRC tests were performed. The first one was used to determine the training intensity; thus, it was done before the start of the aerobic exercise training protocol. The second one was performed at the end of the experimental period in order to verify the effectiveness of the exercise training protocol. The maximum speed (m/min) achieved was recorded.

#### 2.2.2. Aerobic Exercise Training (AET)

This AET protocol has been successfully used by our group to study cardiac and skeletal muscle adaptations for more than a decade [22,23,24,25,26,27,28]. It was performed in a motor treadmill and consisted of 60 min session, 5 days per week, at 60% of the maximum speed achieved in the first MRC test, which corresponds to the maximal lactate steady state. To determine whether AET could prevent cancer-associated cachexia, the protocol was performed for 43 days; thus, mice were trained for 30 days before CT26 cell injections and for 13 more days after the injection. To verify the effectiveness of the AET protocol, the exercise tolerance was evaluated through MRC test after the AET period.

#### 2.2.3. Single Exercise Session

A single exercise session to evaluate molecular changes induced by cancer cachexia was also performed. Therefore, different groups of healthy and CT26 tumor-bearing mice were submitted to a single exercise session, which consisted of the MRC test described above, as previously performed by our group [25]. Mouse groups were killed 0, 3, 6, 12, or 24 h after the session and were compared with the control non-exercised group (termed rest). The maximum speed (m/min) achieved in the MRC test was recorded and used as performance index.

### 2.3. Adult Skeletal Muscle In Vivo Electrotransfer and Plasmids Constructs

Mice were deeply anaesthetized by an intraperitoneal injection of 85 mg/kg ketamine and 10 mg/kg xylazine. A minor incision was done on hindlimb to expose the tibialis anterior (TA) muscle, which was injected along its length with 30 μL of 0.9% saline containing purified plasmid DNA (PureLink HiPure Plasmid Filter Purification Kit, Invitrogen, Carlsbad, CA, USA). To evaluate the hypertrophic capacity of the cachectic muscles, 15 μg of a plasmid coding for a constitutively active form of Akt protein (myr-Akt) or p70 ribosomal S6 kinase 1 (S6K1) were injected into muscle. In order to monitor the gene delivery in the electrotransfer experiments, a plasmid encoding green fluorescent protein (Snap-GFP) was co-transfected into the muscle. Thereafter, two stainless steel spatula electrodes were placed at each side of the isolated TA muscle belly, and electric pulses were applied using an electroporator (AVS Projetos). Five square-wave pulses with a pulse length of 20 and 200 ms intervals between each pulse were delivered at 21 volts [29]. Muscle damage was minimal, and myofibers with abnormal morphology were excluded from the analysis. The myr-Akt plasmid was purchased from AddGene (Addgene plasmid #9008; http://n2t.net/addgene:9008, accessed on 11 January 2018). S6K1 and GFP plasmids were a gift from Dr. Bert Blaauw (University of Padua, Padua, Italy).

### 2.4. Muscle Morphology and Quantitative Analyses

After being collected, TA muscles were immediately frozen in liquid nitrogen. Samples were cut into 7 µm cross sections on a cryostat (CM3050; Leica, Wetzlar, Germany). To reveal the overall morphology, unfixed histological sections were stained with hematoxylin and eosin (H&E) and then analyzed under a light microscope equipped with a digital charge-coupled device camera. The morphometric and quantitative analyses were performed using the computer software SMASH—Semiautomatic Image Processing of Skeletal Muscle Histology [30]. To assess the myofiber cross-sectional area (CSA), a total of approximately 500 myofibers were measured. Figures were mounted using Adobe PhotoShop CS6 (Adobe Systems Inc., San Jose, CA, USA), with image manipulation being restricted to overall threshold and brightness adjustments, which were applied identically to all images.

### 2.5. Immunostaining

Muscle cross-sections to be used for immunodetection were fixed with 4% paraformaldehyde in 0.2 M phosphate buffer (PB) for 10 min at room temperature, blocked with 0.1 glycine in phosphate-buffered saline (PBS) for 5 min, and permeabilized in 0.2% Triton X-100/PBS for 10 min. The slides were then incubated overnight in a moisture chamber at 4 °C with a solution containing the primary antibodies together with 3% normal goat serum and 0.3% Triton X-100/0.1 M PB. Thereafter, the slides were washed (three 10 min washes with 0.1 M PB), and a solution containing the respective secondary antibodies and 0.3% Triton X-100/0.1 M PB was added. The slides were then maintained in this solution for 1 h in a dark box at 37 °C. Finally, the slides were again washed in 0.1 M PB (three 10 min washes), after which they were cover-slipped. The stained sections were analyzed in a microscope (PCM2000, Nikon, Melville, NY, USA). Figures were mounted using Adobe PhotoShop CS6 (Adobe Systems Inc.) with image manipulation being restricted to overall threshold and brightness adjustments.

### 2.6. Western Blot Analyses

TA muscles samples were homogenized in RIPA buffer, composed of 150 mM NaCl, 50 mM Tris-HCl (pH 8.0), 0.1% sodium dodecyl sulphate, 1% NP-40 plus proteinase inhibitor cocktail (1:100; Sigma-Aldrich, St. Louis, MO, USA), and phosphatase inhibitor cocktail (PhosSTOP 1:10; Roche, Basel, Switzerland). To determine the amount of ubiquitinated proteins and the expression of 20S proteasome, samples were homogenized in an extraction solubilization buffer, composed of 50 mM Tris (pH 7.5). Homogenates were then centrifuged at 12,000× *g* for 10 min at 4 °C, and total protein was quantified. Equal amounts of protein (40 mg) were electrophoresed on pre-casted polyacrylamide gels (4–15%, MiniProtean TGX, Bio-Rad, Berkeley, CA, USA) and transferred to a nitrocellulose membrane (Bio-Rad). Membranes were incubated overnight at 4 °C with primary antibodies. Detection of the labelled proteins was achieved using the Odyssey^®^ Fc Imaging System (LI-COR; Lincoln, NE, USA).

### 2.7. Antibodies for Western Blot and Immunostaining

The following antibodies were used for Western Blot experiments: Akt, phospho-Akt (S473), 4E-BP1, phospho-4E-BP1 (T37/46), eIF-4E, eIF-2α, phospho-eIF-2α (S51), and GAPDH. All of them were used at 1:1000 dilution and were from Cell Signalling Technology (Danvers, MA, USA). S6 and phospho-S6 (S240/244) were used at 1:1000 dilution and were from CUSABIO Technology LLC (Houston, TX, USA). A mouse polyclonal antibody raised against ubiquitin (1:200) and anti-20S proteasome core subunits (1:1000) were also used, both from Calbiochem (San Diego, CA, USA). For all of them, the secondary antibody used was the IRDye 800CW (1:15,000 LI-COR; Lincoln, NE, USA). In addition, an anti-puromycin monoclonal antibody was also used (1:1000, Millipore, Burlington, MA, USA). The secondary antibody used in this case was Peroxidase AffiniPure Goat Anti-mouse Fcy subclass 2a specific (1:10,000; Jackson ImmunoResearch Laboratories Inc., West Grove, PA, USA).

The following antibodies were used for immunostaining: rabbit monoclonal anti-laminin (1:200; Sigma-Aldrich, Steinheim, Germany) and rabbit polyclonal anti-GFP (1:200; ThermoFischer Scientific, Waltham, MA, USA). The following secondary antibodies were used: goat anti-rabbit Alexa Fluor 488 and goat anti-rabbit Alexa Flour 568 (1:200; ThermoFischer Scientific).

### 2.8. Muscle Mechanics

#### 2.8.1. In Vivo

Grip strength was measured in mice using a Digital Grip Strength Meter (Bonther Equipments, Ribeirão Preto, Brazil). Three measurements were taken from each mouse, and the force was calculated as the average of the measurements and expressed in millinewtons (mN). This test was performed in the healthy and tumor-bearing mice groups studied in the AET protocol to verify its effectiveness in preventing the loss of muscle force induced by cancer cachexia.

#### 2.8.2. In Situ

Force measurements were performed as described previously [31,32]. Briefly, mice were anaesthetized with 2,2,2-Tribromoethanol (Sigma-Aldrich, Steinheim, Germany, 1 mL/100 g body weight, i.p.). An incision was made on the ankle to expose the TA muscle tendon, and a lateral incision on the thigh was made to expose the sciatic nerve, in which a platinum electrode was connected. The hind limbs were then fixed, and the TA tendon was connected to a force transducer (Biopac Systems, Goleta, CA, USA). Muscle force was recorded and analyzed using the AcqKnowledge System program, version 3.9.1.6 (Biopac Systems). Mice were submitted to external warming in order to maintain core temperature throughout the procedure. At the start of the experiment, the muscle was set to the optimum length (L0, defined as the length resulting in maximum twitch strength). Therefore, the force was determined by stepwise-increasing stimulation frequencies of 4, 20, 55, 75, 100, and 150 Hz, pausing for 30 s between stimuli to avoid effects due to fatigue [33]. Muscle force at 150 Hz was analyzed and absolutely expressed in mN, as well as normalized by the TA muscle mass (mN/g).

### 2.9. Measurements of In Vivo Protein Synthesis

Skeletal muscle protein synthesis was measured in vivo by the SUnSET method as previously described [34]. Briefly, thirty minutes before the muscles were excised, an intraperitoneal injection of 0.04 µmol/g puromycin (Gibco, ThermoFischer Scientific, Waltham, MA, USA) dissolved in 100 µL of physiological solution (NaCl, 0.9%) was given to the animals. Thereafter, puromycin expression was analyzed by Western blot.

### 2.10. Proteasome Activity

The activity of the 20S proteasome was quantified in the cytosolic extract of muscles. Samples were disrupted in lysis buffer (50 mM Tris, pH 7.5; Sigma, St. Loius, MO, USA). Extracts were centrifuged (10,000× *g*, 10 min, 4 °C), and protein concentration was determined. Then, 25 µg of protein from each sample was incubated with 2.5 mM of the fluorogenic proteasome substrates (Calbiochem, Darmstadt, Germany), which are specific for the quantification of chymotripsin-like activity (suc-LLVY-AMC). The increase in fluorescence during substrate cleavage by the proteasome was monitored for 45 min in a fluorescence spectrophotometer, with excitation at 356 nm and emission at 440 nm (Molecular Devices, Sunnyvale, CA, USA).

### 2.11. Statistics

The data are presented as the mean ± standard deviation. Studies consisting of two groups were analyzed using unpaired Student’s *t*-test. Multiple comparisons were made using one-way ANOVA or two-way ANOVA followed by a Tukey *post hoc* test. Significance was defined as a *p* value of less than 0.05 (95% confidence).

## 3. Results

### 3.1. Murine Colon Adenocarcinoma Cell Line, CT26, Induces Severe Cachexia in Mice

Colon adenocarcinoma cells have been used as a model to study cancer-induced cachexia in mice [35]; however, before starting the current investigation, it was firstly necessary to determine the cachectic potential of those cells in our mice. Therefore, 10^6^ cells were subcutaneously injected in the right flank of Balb/C mice, and afterwards, mice were followed up and had their body mass, tumor growth, and overall conditions evaluated every other day for 17 days (Figure 1A). A progressive decrease in the body mass of the tumor-bearing mice (CT26 group) was observed starting at 9 days after the cell injections and becoming significant on days 11 (11%), 13 (17%), 15 (20%), and 17 (21%) when compared to the healthy control non-injected mice (CTRL group) at the same time points (Figure 1B). In the CT26 group, a significant loss of body mass was observed on days 15 (10%) and 17 (12%) post-injection when compared to day 1 post-injection (Figure 1B). In fact, a huge cancer-associated body mass loss was observed at day 17 of the experimental protocol (Figure 1C).

The tumor growth curve (Figure 1D) shows that its rising was proportional to the decrease of body mass observed, since this curve started rising steeply from day 9 post-injection forward, which was the same time point when body mass started its progressive decrease (Figure 1B). Because the main aim of this study was to better understand the skeletal muscle plasticity during cancer cachexia development, CT26-bearing mice were followed up until they presented with a significant body mass loss without signs of ulcerations in the tumor mass, thus avoiding any kind of tumor-burden-associated suffering. Therefore, the longest endpoint established in this study was 17 days post-injection, at which the tumor volume reached around 1200 mm^3^ (Figure 1D) without signs of ulceration, and a significant body mass loss was observed (Figure 1B,C). It is noteworthy that at 17 days post-injection, no mice had died. Thereafter, the fat mass and muscle mass content of both groups were assessed. A huge decrease in the fat mass content (95%, Figure 1E) and a significant reduction in gastrocnemius and TA muscles mass in CT26 group were observed 17 days after cell injections when compared to the CTRL group (35%, Figure 1F–H).

As all the subsequent experiments were performed on TA muscles, histological cross-sections of those samples from mice examined on day 17 post-injection were stained with H&E and then microscopically analyzed (Figure 1I). TA muscle samples from the CTRL group exhibited polygonal myofibers and a normal tissue structure. On the other hand, the cachectic TA muscle displayed clear signs of atrophy, as evidenced by a significantly smaller CSA when compared with the control (39%, Figure 1J,K). In addition, the cachectic TA muscles presented signals of constant damage/regeneration cycles, as indicated by split myofibers and those with centralized nuclei (Figure 1I).

The quantity of skeletal muscle mass is regulated by the net interplay between protein synthesis and degradation. In this sense, the activity levels of the ubiquitin proteasome system (UPS) can be used as a tool for monitoring the loss of muscle mass. Therefore, we decided to first analyze the amount of ubiquitinated proteins; however, differences between both CTRL and CT26 groups were not observed at 17 days post-injection (Appendix A). In addition, the expression of the 20S proteasome subunit was analyzed, and no difference was found between both groups (Appendix A). Finally, the activity of the proteasomal catalytic unit 20S was evaluated in its most activated site of degradation (LLVY), and a slight but non-significant decrease (13%) in the proteasome activity was observed in the CT26 group when compared to the CTRL group (Appendix A).

Therefore, besides showing that CT26-tumor bearing mouse is a robust model to study cachexia, these results indicate that, at late stages, this model does not present significant changes on the proteolytic status represented by the ubiquitin-proteasome system. Hence, we have decided to focus our attention on the anabolic profile for the following analyses.

### 3.2. Aerobic Exercise Training Delays Tumor Progression, Prevents Cachexia, Modulates Akt/mTORC1 Signaling, and Suggests Eukaryotic Initiation Factor-2α as a Target Involved in the Skeletal Muscle Plasticity in Cancer Context

AET has been largely recognized as a non-pharmacological intervention preventing or delaying the progression of many chronic non-transmissible diseases, such as heart failure and diabetes [36]. In addition, it has been suggested that AET can be a complementary therapy for cancer cases [37]. However, because the impact of the AET on skeletal muscle mass maintenance during cancer progression is still not well documented, we designed an approach to evaluate the preventive effects of AET in our cancer cachexia model. The experimental protocol was conducted over 43 days, and during the first 30 days of intervention two groups were used: one remained untrained (termed CTRL group) and the second group underwent the AET protocol (termed AET group). On day 30, the CT26 tumor cells were injected in a subset of each of the two groups; that is, the CTRL and AET groups were split into another two groups: the untrained one injected with CT26 tumor cells (termed CT26 group) and the trained one injected with CT26 tumor cells (termed CT26 + AET). As the AET was maintained for an additional 13 days, the AET and the CT26 + AET groups were trained for 43 days. In the same sense, the CT26 untrained group was evaluated after 13 days of CT26 cell injections (Figure 2A). Firstly, it was observed that the CT26 group presented with a significant decrease in its body mass (13%, Figure 2B) and in its TA muscle mass (14%, Figure 2C) when compared to the CTRL group. On the other hand, the effectiveness of the AET protocol in preventing cachexia was clear because neither a decrease in body mass nor a decrease in TA muscle mass was observed in the CT26 + AET group when compared to the CTRL group (Figure 2B,C, respectively). This protection against muscle wasting induced by AET was reflected in both physical tests analyzed. It was observed that mice from the CT26 group had a worse performance in the maximum running capacity (MRC) test when compared to the CTRL group (37%, Figure 2D). The performance in the CT26 + AET group was higher than that observed in the CT26 group (47%, Figure 2D); however, it was still significantly lower than the CTRL group (8%, Figure 2D). Although MRC is a specific physical test which evaluates both muscular and cardiovascular adaptations to the exercise, we also tested the overall mice capacity in force production, and thus a grip strength test was performed. Mice from the CT26 group presented with a clear decrease in force production in comparison to the CTRL group (25%, Figure 2E); on the other hand, mice from the CT26 + AET group could generate the same force levels observed by mice from the CTRL group (Figure 2E). Besides this, being in part due to muscle mass maintenance, those preventive effects induced by AET were also causally linked with a delay in tumor growth, because slower tumor progression was obvious in the CT26 + AET group (Figure 2F).

Because it was recently reported by our group that eukaryotic initiation factors, specially the eIF-2 family and the ribosomal proteins, were downregulated elements identified in cachectic muscles [12], we questioned whether such prevention in the muscle mass loss induced by AET could be related to modulations in the hypertrophy signaling. Since it is well established that one of the major pathways regulating adult muscle mass is the Akt/mTORC1 pathway, which is thought to act mainly through increases in protein synthesis by modulating translation initiation [15,16], we firstly analyzed the phosphorylation levels of Akt protein, which was increased in CT26 when compared with the CTRL group (Figure 2G,H). Interestingly, the long-term AET, started before the tumor development, was able to keep the Akt phosphorylation at the same levels of the non-tumor bearing CTRL group, thus suggesting that AET can modulate the dysregulation of the main controller of the pathway. No change in the phosphorylation levels of the ribosomal S6 protein, a direct target of mTORC1, was observed when the CT26 group and the CTRL were compared (Figure 2G,I). However, the CT26 + AET group displayed a decrease in those levels in comparison to both the CTRL and CT26 groups (Figure 2G,I), which can be explained by the effects generated by those lower levels of Akt in the CT26 + AET group. The expression of the 4E-BP1 binding protein was also evaluated, but no significant change in its levels among the groups studied were found (Figure 2G,J). Finally, a significant increase was observed in the phosphorylation levels of eIF-2α in the CT26 group in comparison with the CTRL group, which indicates suppression in the muscle protein synthesis (Figure 2G,K). Interestingly, the AET applied in the CT26 + AET group was able to maintain the phosphorylation of eIF-2α at the same levels of both non-tumor-bearing groups.

In order to better understand the intrinsic ability of cachectic muscle to activate protein synthesis pathways induced by a single exercise session, we used two subsets of mice, one not injected with the tumor cells (termed CTRL) and a second one injected with the tumor cells (termed CT26 group). The levels of key proteins involved in synthesis pathways were evaluated in both groups over different recovery time points after the single session of exercise. Thirteen days after the injections (i.e., the same time point used in the previous AET approach), mice were randomly divided into six different groups. One of those six groups did not perform the single exercise session, and thus both CTRL and CT26 were termed *rest*. The other five groups performed the MRC test, which was used as the single exercise session as previously reported by our group [25], and were killed at different time points: immediately after (0 h), and 3 h, 6 h, 12 h, or 24 h after the single exercise session (Appendix A). TA muscle mass loss confirmed that our cachexia model induced atrophy in all the six groups (Appendix A), and due to this muscle wasting, the five CT26 groups that underwent the single exercise session had a worse physical performance in the MRC test, since the maximum speeds achieved by them were lower (~25% to 35%) than those from their respective CTRL groups (Appendix A). We then used TA muscle samples to perform Western blot experiments, and an increase in the phosphorylation levels of Akt protein was first observed in the CT26 group at *rest* when compared to their respective CTRL group (Appendix A), suggesting a compensatory response to muscle wasting, and the single exercise session did not significantly alter those levels at all time points. When the phosphorylation levels of the ribosomal protein S6 were evaluated, the unique significant change observed was between CTRL and CT26 groups analyzed at 24 h after the single session (Appendix A). In addition, no significant change was observed in the phosphorylation levels of the 4E-BP1 binding protein, except for the decrease observed in the CTRL group analyzed immediately after the acute session (i.e., at 0 h time point) (Appendix A). Corroborating our previous AET approach, we observed increased phosphorylation levels of the eIF-2α in the CT26 group at *rest* in comparison with the respective CTRL group (Appendix A), thus reinforcing the possible dysfunction in the translation initiation process displayed by cachectic muscles. An apparent normalization in those levels was observed in the CT26 group at 0 h, 3 h, 6 h, 12 h, and 24 h after the single exercise session when compared to the *rest* condition (Appendix A). In contrast, an increase in those phosphorylation levels on the CTRL group was observed at 3 h and 6 h after the single exercise session, which can be explained by the protein turnover suggested by proteolysis induced after a single bout of exercise. Those levels were also normalized at 12 h and 24 h after the single exercise session.

Collectively, these results indicated that the aerobic physical exercise, when applied previously to the cancer development, delayed cachexia progression and indicated that the mechanism controlling muscle mass in the context of cancer is governed by the translation initiation functionality.

### 3.3. Akt-Induced Hypertrophy Is Necessary to Prevent and to Revert Cancer-Associated Muscle Wasting

It has been suggested that ribosome biogenesis is necessary to induce muscle hypertrophy [38] and to generate muscle force [39], and since we have previously identified altered ribosome protein in a proteomic screening in cachectic muscles [12], we firstly questioned whether an in vivo overexpression of ribosome proteins would revert such alteration and counteract muscle wasting, thus being a possible target to treat cancer cachexia. Therefore, we electroporated a plasmid coding for a constitutively active form of p70 ribosomal S6 kinase 1 (S6K1) in the TA muscles of CT26-bearing mice. S6K1 was chosen by the fact that with mTORC1-induced increased S6 activity, there is an increase in mRNA biogenesis and the translation of ribosomal proteins [40]. For this approach, a subset of mice was subcutaneously injected with CT26 cells, and on the same day, electroporation with S6K1 was performed to verify whether the ribosomal protein activation would prevent muscle wasting (Appendix A). TA muscles from the right hind limb were electroporated with the S6K1 plasmid, and the TA muscles from the left hind limb were electroporated with an empty vector. Ten days later, a time point at which the mice started to present signs of cachexia (see Figure 1), the muscles were collected and microscopically analyzed. Since we did not observe any difference in the myofiber cross-sectional area induced by the empty vector co-transfected with the Snap-GFP plasmid (Appendix A), we analyzed the S6K1 effect on the right TA from the CT26-bearing mice, confronting the positive green transfected myofibers (CT26 S6K1 +) with the surrounding non-transfected ones (CT26 S6K1 −), as previously reported [18,39]. However, no differences in the myofiber CSA were observed (Appendix A). Thereafter, we questioned whether the electroporation with S6K1 would be capable of reverting muscle wasting once cachexia was already established. Therefore, the S6K1 plasmid was electroporated 10 days after the CT26-cell injections, and the TA muscles were analyzed seven days later (Appendix A). Once again, no differences were observed in the myofiber CSA (Appendix A). These negative results can be explained by the fact that the electroporation with S6K1 plasmid was not sufficient to generate a hypertrophic signal to counteract cachexia as AET did (see Figure 2 and Appendix A), because to display its functions S6K1 needs to be activated by phosphorylation from high-level stimuli into the Akt/mTORC1 pathway. Likewise, no differences were observed in the myofiber CSA (Appendix A).

These negative results might be associated with an impaired ability of S6K1 plasmid electroporated muscles to generate a hypertrophic signal to counteract cachexia when compared with AET (see Figure 2 and Appendix A). In fact, the muscle mechanical stimuli by AET induces high levels of Akt/mTORC1 pathway activation, fulfilling S6K1 phosphorylation. To overcome this limitation, we tested the electroporation with a plasmid coding for a constitutively active form of Akt (myr-Akt) in cachectic TA muscles, which certainly induces a higher stimulation of the Akt/mTORC1 pathway. In fact, it was previously shown that in vivo activation of Akt can induce myofiber hypertrophy in regenerating denervated muscles [18]; however, it had still not been tested in cancer-induced muscle wasting. Hence, to verify if the Akt activation could prevent cancer cachexia, mice were subcutaneously injected with CT26 cells on the same day of the electrotransfer experiments, and ten days later, muscles were collected and analyzed (Figure 3A). It was observed that the CSA of the Akt-positive myofibers were significantly bigger than the surrounding non-transfected myofibers (86%, Figure 3B–D). Thereafter, we questioned whether the Akt-induced hypertrophy would be capable of reverting muscle wasting, since cachexia was already established. To elucidate this matter, we performed a therapeutic approach in which the Akt plasmid was electroporated 10 days after the CT26 cell injections, and the TA muscles were analyzed seven days later (Figure 3E). Interestingly, it was also observed that the CSA of the Akt-positive myofibers were significantly bigger than the surrounding non-transfected ones (80%, Figure 3F–H and Appendix A). Therefore, we showed that Akt-induced hypertrophy is able to prevent and, most importantly, to revert cancer-induced muscle wasting, thus highlighting that hypertrophic potential is still preserved in the skeletal muscle tissue even under a severe atrophy scenario.

### 3.4. Akt-Related Hypertrophy Rescues Muscle Force and Induces a Better Translation Initiation Process in Cachectic Muscles

To better understand the physiological and molecular profile underlying the re-established muscle CSA induced by in vivo Akt activation, we analyzed force production and signaling changes in TA samples from the CT26—Akt^+^ group, which were electroporated with Akt plasmid 10 days after the CT26 cell injections and then collected seven days later, by comparison with samples from the CT26 group, also at 17 days post-cell injections, and with samples from a healthy control group (CTRL). To verify the capacity of the cachectic muscles to produce force and the effects of the electroporation with Akt on that, it was necessary to perform a specific strength test, and thus we chose to use the in situ mechanical assay, in which the TA muscle and its tendon could be isolated. As expected, the cancer-related muscle wasting significantly reduced both absolute (51%) and relative force (19%) as observed in the CT26 group when compared with the CTRL group (Figure 4A,B). The electroporation with Akt induced a slight increase (20%) in the levels of absolute force generated by the CT26–Akt^+^ group when compared with the CT26 group (Figure 4A). Surprisingly, when the relative force was evaluated, Akt electroporation was able to normalize the relative muscle force by the same average found in the CTRL group (Figure 4B), besides not affecting muscle weight (CT26: 34 ± 3 mg; CT26–Akt^+^: 36 ± 1 mg).

Next, to determine whether the electroporation with Akt would increase the rate of muscle protein synthesis, we performed the SUnSET assay. A huge increase in the puromycin incorporation in the CT26–Akt^+^ group was observed in comparison to the CT26 group, thus suggesting that in vivo Akt activation was able to increase protein synthesis in cachectic muscles (Figure 4C). Western blot analyses were also performed to evaluate modulations in the Akt/mTORC1 signaling pathway and the expression of eukaryotic initiation factors. Differently from the previous analyses, in which we did not find any changes in the phosphorylation levels of S6 protein between cachectic and non-cachectic muscles, herein, at 17 days post-injection lower S6 levels were observed in the CT26 group than in the CTRL group, which were restored by Akt-induced hypertrophy (Figure 4D,E). Corroborating previous experiments, we did not find any change in the phosphorylation levels of the 4E-BP1 binding protein in any of the groups analyzed (Figure 4D,F). In contrast, increased phosphorylation levels of the eIF-2α in the CT26 group were observed in comparison to the CTRL group, and those levels were normalized in the CT26–Akt^+^ group by the in vivo Akt activation (Figure 4D,G). In addition, we verified the expression of eIF-4E, a eukaryotic initiation factor which is one of the targets of mTORC1 signaling. It was found that eIF-4E expression levels were lower in the CT26 group, thus corroborating the hypothesis of dysfunction in the translation initiation, which was reverted by the Akt stimulation (Figure 4D,H).

## 4. Discussion

Besides being responsible for up to 30% of deaths in cancer cases, cachexia is considered a serious public health problem since it reduces tolerability and response to treatments, which contributes to increases in hospitalization costs. Since there is no approved treatment for cancer cachexia at present, an understanding of the mechanisms that control skeletal muscle mass and function in the context of this syndrome could pave the way for the discovery of innovative therapeutic strategies to counteract muscle wasting.

It has already been demonstrated that AET induces organisms’ defenses against the development of different types of tumors, as well as the incidence of metastasis [10]; however, the effects of AET in preventing cancer-related muscle wasting are still poorly understood. In this sense, one of the major findings of the present study is that AET in tumor-bearing mice, when started previously to the cancer development, is safe and delays the tumor growth, preventing muscle mass loss and dysfunction. Indeed, the main responsible mechanisms underlying this response is related to modulations in the Akt/mTORC1 signaling pathway, besides reducing the eIF-2α expression levels.

As is widely known, one of the major signaling pathways regulating adult muscle mass is Akt/mTORC1 [41], which is indeed recognized to act mainly through increases in protein synthesis by modulating translation initiation [15,16]. In addition, eIF2 is a GTPase that forms a ternary complex with GTP and Met-tRNA and assembles with the 40S ribosomal subunit, forming the 43S pre-initiation complex. It is already recognized that phosphorylation of the alpha subunit of the eIF2 at serine 51 suppress translation initiation [42].

Our group have demonstrated that AET induces benefits to the skeletal muscle tissue in heart-failure-induced muscle wasting, and some of those benefits are linked to modulations in Akt/mTORC1 signaling [23]. In another interesting work, in which a colon cancer mouse model of cachexia was also used, the effects of both aerobic and resistance training were compared [43]. Both protocols were unable to prevent tumor-induced body weight loss; however, the study attributed therapeutic value only to AET, since it marginally rescued muscle mass and preserved function through restoration of the expression levels of mTOR. Such different outcomes observed between that study and ours might be due to the different exercise protocols used (i.e., session duration and intensity), which were higher in the present. Thus, identifying the most adequate exercise protocol to prevent and to treat cancer cachexia is a necessary step in future investigations.

Besides preventing muscle loss and dysfunction, our AET protocol also preserved running capacity, which is a physical performance index that is affected by both cardiopulmonary and muscular adaptations. These responses can be in part attributed to the recently described role of mTORC1 in regulating neuromuscular junction stability [44], and also by the recent finding of our group which showed that AET is capable of delaying cardiac remodeling in the same model of cancer cachexia. One might speculate that the maintenance of running capacity might be also linked to improved respiratory function. In this sense, it is known that diaphragm muscle atrophy and weakness exacerbate symptoms of breathlessness and impair ventilation, which can lead to life-threatening respiratory failure in cancer patients [45]. However, since the molecular basis of diaphragm weakness in cancer remains poorly understood, future investigations will be important to explore the diaphragm adaptations, and to improve knowledge about the therapeutic effects of AET in cancer cases.

Although reinforcing the concept that exercise training is a safe complementary therapy for cancer, it is important to point out that in our study, exercise had started before the injection of the tumor cells when mice were still healthy, which highlights the importance of maintaining regular levels of physical activity in the long term. In fact, a healthy lifestyle reduces the incidence of many types of cancer [46]. We have also observed previously that exercise therapy applied after the injection of tumor cells was able to prolong survival and improve skeletal muscle functionality in a rat model of cancer cachexia [12]. However, one might consider that a significant part of the population worldwide is physically inactive and adopts sedentary behaviors, which calls attention to increasing strategies to combat physical inactivity [47]. In addition, physical activity strategies cannot accommodate all situations, as cancer patients often undergo surgeries and/or long-term chemotherapy treatments that preclude participation in regular physical exercise programs (i.e., are often bedridden).

In this sense, the Akt/mTORC1 pathway, due to its complexity, can be modulated in several ways, thus suggesting that there are potential intrinsic targets for therapeutic approaches and drug development to counteract muscle atrophy [17]. Conversely, some types of cancers are treated with mTOR inhibitors to reduce tumor cell proliferation [48]. Even targeted therapeutics of tumors, which do not derive from mTOR mutations, are greatly aided by co-treatment with mTOR inhibitors to delay innate or acquired resistance to the drugs [49]. In such a context, a systemic mTOR inhibitor can be interesting in cancer-related treatments, but considering the key role Akt/mTORC1 signaling plays in muscle physiology and metabolism [50], this kind of drug can be detrimental for the muscles.

Since it was already demonstrated that in vivo Akt-induced hypertrophy could revert muscle wasting in a mouse model of denervation [18], we tested a similar approach in our cancer model, and thus we show for the first time that in vivo Akt/mTORC1 pathway manipulation is able to reinduce hypertrophy in cancer cachexia. Linked to this data is important translational information indicating that skeletal muscle tissue still preserve a hypertrophic potential for regrowth despite being challenged by the most severe atrophic stress. Thereafter, when we analyzed the hypertrophic signaling in depth, and it was clear that the critical mediator in improving cachexia by stimulating muscle growth is the Akt/mTORC1 signaling pathway. An increase in protein synthesis rate was observed, which was a consequence of a positive modulation in that pathway, since an increase in the S6 ribosomal protein phosphorylation rate and restored levels of eukaryotic initiation factors were observed. It is important to note that such a decrease in the phosphorylation levels of S6 protein of cachectic muscles was restricted to mice studied in the Akt protein gain of function approach, and not in those mice that underwent the AET protocol. We attribute such difference to the period of the analysis of both approaches (17 days and 13 days after CT26 cell injections, respectively). Seventeen days after tumor cell injection, the cachectic stage is so advanced that it leads to muscle functional collapse.

It was also observed that the Akt-induced hypertrophy could partially prevent the muscle functional impairments induced by cancer cachexia. The better muscle force production is linked to elevated levels of protein synthesis and can certainly be attributed to the restoration in S6 phosphorylation levels, since it was already reported that activation of ribosome biogenesis is required to maintain muscle force production during hypertrophy [39].

Finally, it was observed that Akt-induced hypertrophy was also able to decrease the eIF-2α phosphorylation rate, thus reverting translation initiation dysfunction in cancer cachexia, as AET did. In fact, a previous study by our group reported that the major downregulated signaling pathway found in the cachectic muscles was eIF2 signaling [12]. It suggests that the muscle wasting is not only due to the inflammatory process that exacerbates the proteolysis, but also by a dysfunction in the translation initiation process, which can preclude muscles from re-synthetizing proteins. Indeed, dysfunctions in that eukaryotic initiation factor family was also found in an experimental model of cachexia induced by heart failure [51] and in muscle samples from cancer patients [52].

The remarkably similar adaptive responses coordinated by mTORC1 and eIF-2α signaling suggest potential crosstalk between these important nutrient sensing pathways. Like the Akt/mTORC1 signaling pathway, the integrated stress response system coordinates several cues regarding the metabolic status of the cell and orchestrates an adaptive response through the eIF-2α. The eIF-2α effector ATF4 increases the expression of 4E-BP1, augmenting its disinhibition in response to mTORC1 inhibition, thus indicating that eIF-2α phosphorylation can also affect mTORC1 function [53]. In addition, in the context of skeletal muscle plasticity, it was recently reported that phosphorylation of eIF-2α is a translational control mechanism regulating muscle stem cell quiescence and self-renewal [54,55].

Taking into consideration that after activation of Akt/mTOR signaling, the cachectic muscles preserves their growth ability, which seems to be due to a better function in the eukaryotic initiation factors, and that up to now no pharmacological agent against chronic disease-related muscle wasting has been identified, future studies should investigate whether and how recently described muscle-specific chemical compounds [51,56] could be possible alternatives to counteract cancer cachexia. Overall, our results reinforce the importance of physical exercise to avoid chronic disease-related comorbidities and show for the first time that the Akt-induced hypertrophy can revert cancer cachexia.

## 5. Conclusions

Advances in therapeutic approaches have significantly improved the survival of cancer patients; however, associated factors such as skeletal muscle mass loss and dysfunction still do not have a specific treatment. In the present study, we confirmed that AET is an important adjuvant therapy to treat cancer cachexia. These benefits were related to prevention of muscle mass loss and dysfunction, and the possible mechanisms are linked with modulations in the Akt/mTORC1 pathway, in addition to restoring eukaryotic initiation factors functionality. In addition, we showed for the first time that skeletal muscle tissue still preserves hypertrophic potential for growth even under a severe atrophic scenario, since it was observed that the in vivo activation of Akt protein was able to revert cancer cachexia (see Figure 5). Together, these results increase understanding of the mechanisms that control skeletal muscle mass and function in the context of cancer and highlight the relevance of muscle-specific therapeutic strategies to counteract chronic disease-related muscle wasting.

## Figures and Tables

**Figure 1 cancers-14-00028-f001:**
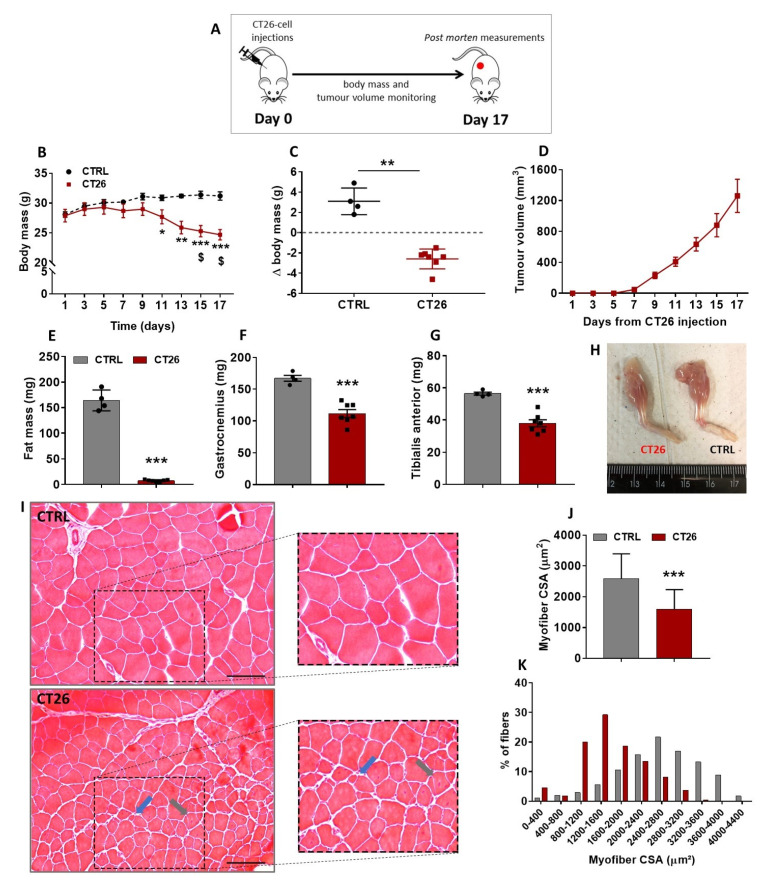
CT26 adenocarcinoma cells induces cachexia in mice. (**A**) Experimental design used to evaluate the cachectic potential of the CT26 cells. Mice were injected subcutaneously with 10^6^ cells and followed up for 17 days. (**B**) Body mass evolution throughout the experimental protocol. It was observed that the CT26-bearing mice presented a progressive and significant loss of body mass from day 9 post-injection forward. (**C**) Body mass changes in CT26-bearing mice in comparison with the healthy control ones at day 17 of the experimental protocol. (**D**) Tumor volume evolution throughout the experimental protocol. Note that the curve starts to rise exponentially from day 9 post-injection forward, which is the same time point in which the body mass started to decrease progressively. (**E**) Total fat mass measured in the end of the protocol. (**F**,**G**) Gastrocnemius and TA muscles mass, respectively, weighed at the end of the protocol. (**H**) Representative photos comparing the loss of muscle mass on hind limbs of both experimental groups. (**I**) Histological features of TA muscle cross sections analyzed in the end of the protocol. The CTRL group presented a normal tissue structure; in contrast, the CT26 group presented smaller myofibers. In addition, cachectic muscle tissue presented split myofibers (indicated by blue arrows) and myofibers with centralized nuclei (indicated by grey arrows). Scale bar: 100µm. (**J**,**K**) Myofiber CSA quantification. The TA muscles from CT26 group presented smaller myofiber cross-sections, and the number of small myofibers is bigger than that of the CTRL group. Data are presented as mean ± SD. * *p* < 0.05 vs. CTRL; ** *p* < 0.01 vs. CTRL; *** *p* < 0.001 vs. CTRL; $ *p* < 0.05 vs. same group at day 1 post-injection. CTRL, n = 4. CT26, n = 7.

**Figure 2 cancers-14-00028-f002:**
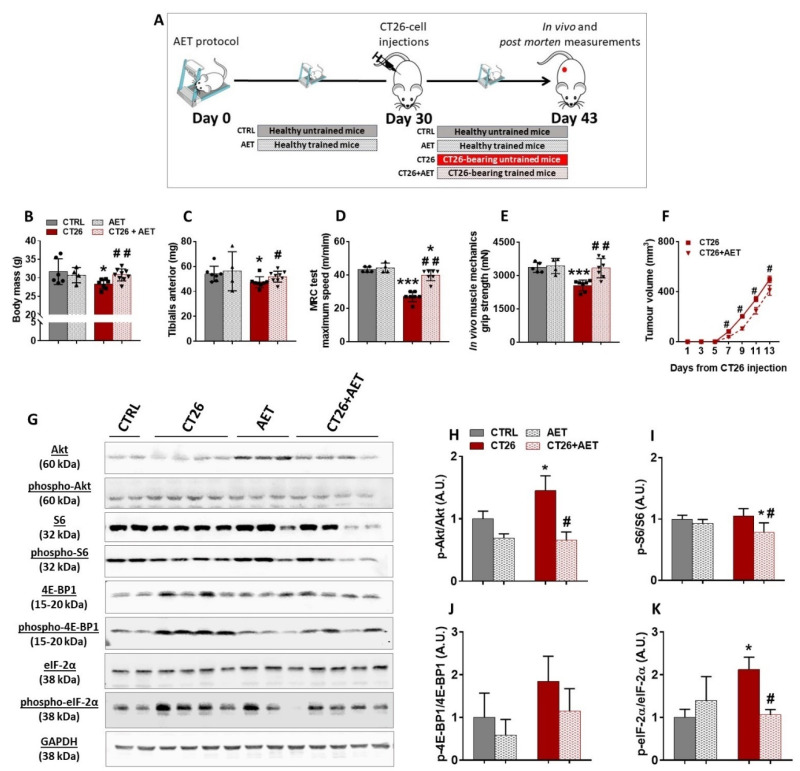
Aerobic exercise training delays cancer cachexia progression. (**A**) Experimental design used to study the preventive effects of the AET in the CT26 cancer cachexia model. Mice were previously trained for 30 days, and then they were injected subcutaneously with 10^6^ CT26 cells. Thereafter, the AET protocol was maintained for 13 more days. (**B**) Body mass analyzed at the end of the protocol. It was observed that the CT26-bearing mice presented with a decrease in body mass, and AET was effective in preventing this loss. (**C**) TA muscles mass measured at the end of the protocol. It is clear to note that AET was able to prevent cancer-related muscle wasting. (**D**) Maximum speed achieved in the MRC test performed at the end of the protocol. The CT26 group had a worse performance, and the AET was able to prevent its drops. (**E**) Maximum force evaluated by the grip strength test at the end of the protocol. A decrease in the force production by the CT26 group, which was prevented by the AET protocol, was observed. (**F**) Tumor volume evolution throughout the experimental protocol. The AET was able to delay the tumor progression, and also contributed to the benefits observed in the muscle tissue. (**G**) Representative Western blot images with experimental groups identified at the top. CTRL: healthy untrained group; CT26: tumor-bearing untrained group analyzed 13 days after the injections; AET: healthy group trained along 43 days; CT26 + AET: tumor-bearing mice group trained over 43 days, which were injected with the CT26 cells at 30 days. Original blots see Appendix A.(**H**–**J**) Protein densitometry analyses of components of Akt/mTORC1 signaling (Akt, phospho-Akt^ser473^, S6, phospho-S6^ser240/244^, 4E-BP1 and phospho-4E-BP1^thr37/46^). (**K**) Densitometry analyses of eIF-2α and phospho-eIF-2α^ser51^. Data are presented as mean ± SD. * *p* < 0.05 vs. CTRL; *** *p* < 0.001 vs. CTRL. # *p* < 0.05; ## *p* < 0.01 vs. CT26. CTRL, n = 5–6. CT26, n = 7–8. AET, n = 4. CT26 + AET, n = 7–8.

**Figure 3 cancers-14-00028-f003:**
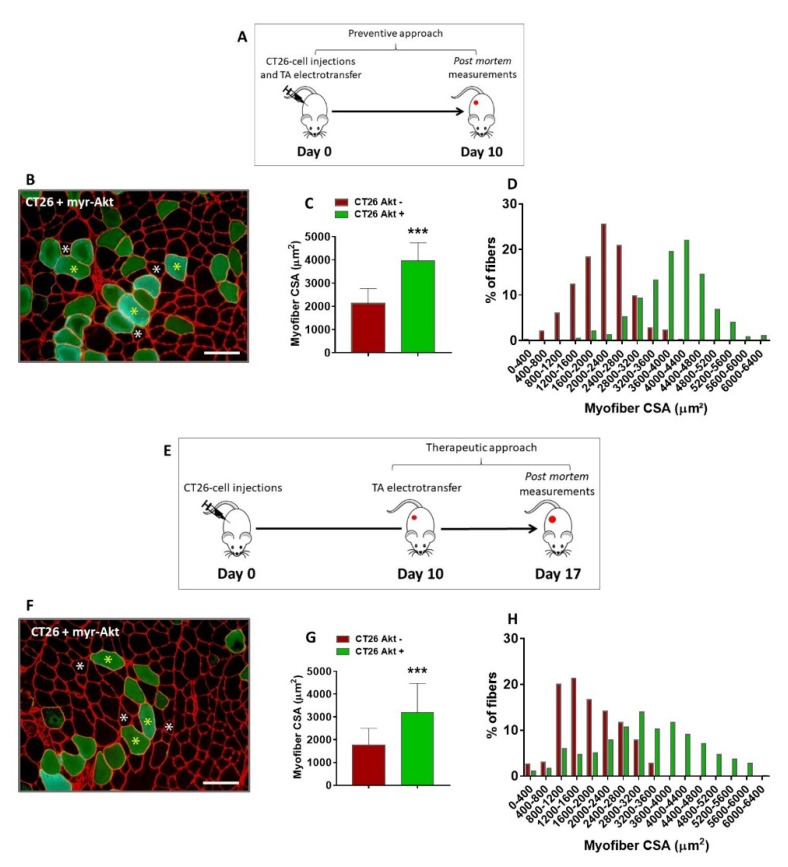
In vivo Akt activation induces hypertrophy and reverts cancer-associated cachexia. (**A**) Experimental design used to determine whether in vivo Akt activation could prevent muscle wasting induced by cancer. Mice were injected subcutaneously with 10^6^ cells and, on the same day, the TA muscles from the right hind limb were electroporated with myr-Akt plasmid. Ten days later, muscles were collected and analyzed. (**B**) TA muscle cross-section from CT26-bearing mouse 10 days after tumor cells injection and myr-Akt electrotransfer immunostained for laminin (red) and GFP (green). Note that the Akt-positive myofibers (green fibers, indicated by yellow asterisks) are bigger than the surrounding non-transfected ones (black fibers, indicated by white asterisks). (**C**,**D**) Myofiber CSA quantification. (**E**) Experimental design used to determine whether in vivo Akt-induced hypertrophy could revert cancer cachexia. Mice were injected subcutaneously with 10^6^ cells, and 10 days later, the TA muscles from the right hind limb were electroporated with myr-Akt plasmid. Seven days later, muscles were collected and analyzed. (**F**) TA muscle cross-section from CT26-bearing mouse at the end of this protocol (i.e., 17 days after tumor cells injection). The cross-sections were immunostained for laminin (red) and GFP (green). The Akt-positive myofibers (green fibers, indicated by yellow asterisks) are bigger than the surrounding non-transfected ones (black fibers, indicated by white asterisks). (**G**,**H**) Myofiber CSA quantification. CT26 Akt *+*: Akt-positive TA myofibers from CT26-bearing mice; CT26 Akt *−*: Akt-negative TA myofibers from CT26-bearing mice. Scale bar: 100 µm. Data are presented as mean ± SD. *** *p* < 0.001 vs. CT26 Akt−. CT26, n = 6.

**Figure 4 cancers-14-00028-f004:**
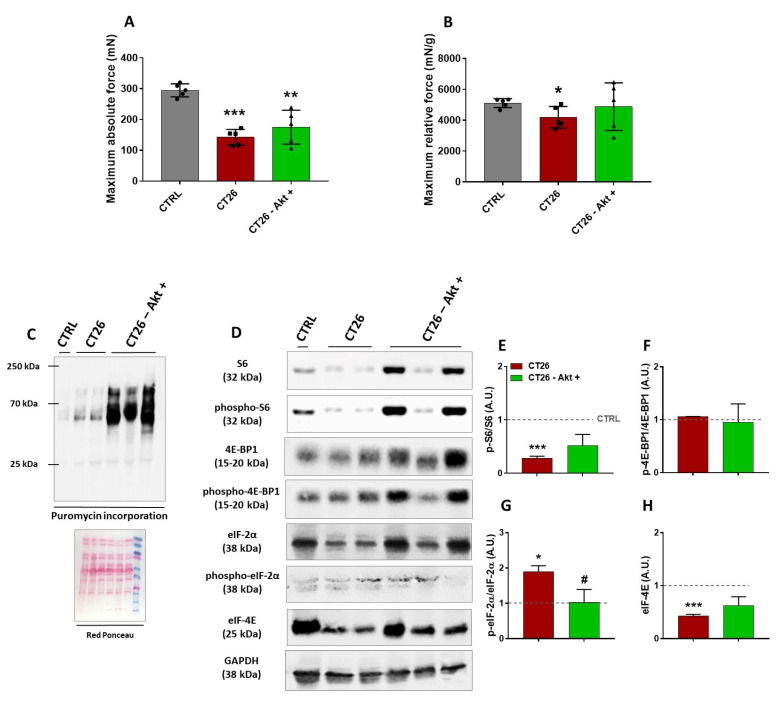
Functional and hypertrophic adaptations induced by the Akt activation in cachectic muscles. (**A**,**B**) Absolute and relative force production analyses, respectively. (**C**) Representative Western blot performed on TA muscles analyzing puromycin incorporation. Experimental groups are identified at the top. CTRL: TA samples from healthy control mice; CT26: TA samples from tumor-bearing mice 17 days after cell injections; CT26–Akt^+^: TA samples electroporated with Akt plasmid 10 days after mice being injected with tumor cells and collected seven days later. Red Ponceau was used as loading control. (**D**) Representative Western blot from the same groups cited above. Original blots see Appendix A.(**E**–**G**) Densitometry analyses of elements of Akt/mTORC1 signaling (S6, phospho-S6^ser240/244^, 4E-BP1 and phospho-4E-BP1^thr37/46^ and eIF-4E). (**H**) Densitometry analyses of eIF-2α and phospho-eIF-2α^ser51^. Data are presented as mean ± SD. * *p* < 0.05; ** *p* < 0.01; *** *p* < 0.001 vs. CTRL. # *p* < 0.05. CTRL, n = 1–5. CT26, n = 2–5. CT26–Akt^+^, n = 3–5. Herein, in some Western blot analyses, a single CTRL sample was used just for reference.

**Figure 5 cancers-14-00028-f005:**
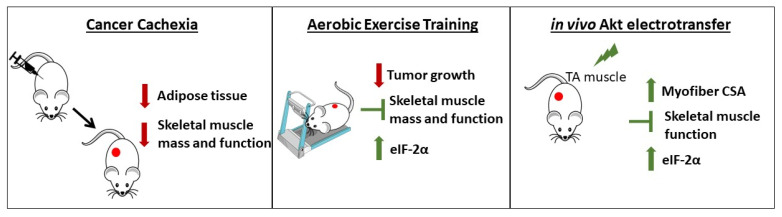
Cancer cachexia-induced alterations and the effects of AET and in vivo Akt electrotransfer. Cancer cachexia induces skeletal muscle mass and function loss leading to a severe atrophic scenario. AET is able to counteract cachexia by both delaying tumor growth and saving the skeletal muscle mass and function. In addition, in vivo Akt stimulation induced myofiber hypertrophy, thus avoiding muscle dysfunction. According to the results presented here, both approaches converge to the same molecular mechanism, which is the increase in eIF-2α expression.

## Data Availability

The datasets analyzed during the current study are available from the corresponding author on reasonable request.

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
