# Peer review of "Aerobic Exercise Training and In Vivo Akt Activation Counteract Cancer Cachexia by Inducing a Hypertrophic Profile through eIF-2α Modulation"

_cancers, 2021, doi:10.3390/cancers14010028_

Round 1

Reviewer 1 Report

The current manuscript aimed at investigating the role of aerobic exercise training and in vivo Akt activation in reducing cancer cahexia as well as understanding the underlying mechanisms. The idea of the study is very novel and original, with a good potential to generate significant added value to current knowledge. 

Introduction is adequate, as it presents the background of the study in a logical manner, pointing out the motivation and need of this research. The cited references are suitable and relevant for the ideas that were presented. 

The study design is complex and very well built, therefore providing sufficient data as to prove the initial hypotheses. Materials and methods are described in sufficient detail as to be well understood and also reproducible. Statistical analysis methods that were chosen are appropriate. 

Result are very well presented, with adequate explanations as well as relevant figures and charts. 

Discussions are very well conducted, point out the originality of the research and provide the reader with several valuable "take-home" messages. The literature cited is adequate. Conclusions are also sound, and in accordance with the results. English language is fine, to the best of my knowledge.

In my opinion, this is a very good quality manuscript, that can be accepted for publication as it is.

Author Response

Answer: We would like to thank you for your help in evaluating this manuscript and trust that this new revised version will be acceptable for publication in the CANCERS.

Reviewer 2 Report

In the given investigation authors have worked on “Aerobic exercise training and in vivo Akt activation counteract cancer cachexia by inducing a hypertrophic profile through eIF-2α modulation” and revealed some exciting findings to understand the mechanism and treatment of cachexia. However, authors have to resolve few minor comments before the final acceptance:

  • The authors have not used new references and there is no reference of 2021. Therefore, inclusion of new references is must.
  • The author should provide a fig explaining mechanism which can explain the benefits of exercise.

Author Response

• The authors have not used new references and there is no reference of 2021. Therefore, inclusion of new references is must.

Answer: We have revised the references and updated the most relative ones, which according to our understanding are those related to cancer cachexia definition, all of them are now from 2020 and can be find highlighted in yellow in this new manuscript version. As our study brings newest data regarding to cachexia, exercise training and Akt overexpression we were not able to find up to date references to match our results. However, we would like to thank the reviewer for this observation that have highlighted to us the necessity to produce a review paper about this current topic.

• The author should provide a fig explaining mechanism which can explain the benefits of exercise.

Answer: We provided a short explaining figure which can be found as Figure number 5 highlighted in yellow in this new manuscript version.

Reviewer 3 Report

The paper titled Aerobic exercise training and in vivo Akt activation counteract cancer cachexia by inducing a hypertrophic profile through eIF-3 2α modulation describes a set of very nice and inter-related studies, which investigate an extremely important topic - cachexia and the mechanism behind skeletal muscle loss.  The paper highlights the important role of the AKT pathway in prevention of muscle loss and suggests aspect of this pathway could be therapeutic targets for muscle atrophy.  This is extremely important because currently there are no really effective treatments for skeletal muscle loss.  The experiments are well designed and nicely illustrated in the figures.  The results are well-presented, thoroughly explained and are supported by the data in the figures.  The methods are appropriate - although I have a question about the electro-transfer (see below).  Over all this is an extremely nice paper, but the English needs improvement.  I made some suggestions early in the paper (see below) and these changes in the sentence structures could be applied throughout the paper.

I do have a question about the Electro-transfer:  GFP was used to ensure transfection occurred – I assume fluorescence was noted.  However, was the group able to quantify this?  It is not discussed.  I ask because I wonder:  How were corrections for differences in electro-transfer efficiency made?  Obviously, one cannot assume that every muscle will “take up” and express the plasmids to the same extent.  Often people will also co-electrotransfer a reporter gene (e.g., luciferase) that is quantifiable and compare this reporter gene expression between muscles as a measure of electrotransfer efficiency.  It is clear in figure 3 that the AKT was expressed because there is a measurable effect that can only be attributed to that plasmid product.  The data show that the expression was likely quite even from animal to animal.  However, in figure S3, it is possible that the negative effect is a result of poor (or uneven) uptake and/or expression of the S6K1 plasmid.  It is my understanding that the GFP was encoded by a separate plasmid that was co-electrotransferred with the plasmid encoding the S6K1.  Would the authors comment on this, please?  Perhaps I am missing something - that is possible.  If so, I apologize.

Also, Figure 1 (line 304):  That looks more like a linear rise than an exponential rise to me….  Both axes appear to be linear.  Do you mean “rise steeply?”

English Suggestions (by line number):

  1. Legends should legend: “…each figure legend.”
  2. Would read better as “At the end of each experimental procedure, ….” Rather than “In the end…”

112 “…in order to verify the….”

117 “…and consisted of 60-minute….”

124-125  “A single exercise session to evaluate molecular changes induced by cancer cachexia was also performed.”

139-140  “…experiments, a plasmid encoding Green fluorescent Protein (Snap-GFP) was co-transfected into the muscle.”

158-159, 172-173  “…to overall threshold and brightness adjustments” - which were applied identically to all images?  

175, 180 “…composed of….”

217  How were the hind limbs “fixed?”

245-246  Would it be better to say “The data are presented as the mean +/- the standard deviation.  Analyses of studies consisting of two groups were analyzed using….”

257-260 I suggest:  “A progressive decrease in the body mass of the tumour-bearing mice (CT26 group) was observed starting at 9 days after the cell injections and becoming significant on days 11 (11%), 13 (17%), 15 (20%) and 17 (21%) when compared to the healthy control non-injected mice (CTRL group) at the same time points (Figure 1B).”

266  Is it a truly exponential rise?

268-271  I suggest:  “Because the main aim of this study was to better understand the skeletal muscle plasticity during cancer cachexia development, CT26-bearing mice were followed up until they presented with a significant body mass loss without signs of ulcerations in the tumour mass, thus avoiding any kind of tumour burden-associated suffering.”

274-275  I suggest:  “It is noteworthy that at 17 days post-injection no mice had died.”

276  I suggest: “Thereafter, the fat mass and muscle mass content of both groups were assessed.”

277:  I suggest a similar sentence structure rearrangement as above.

288-297   “….however, differences between both CTRL and CT26 groups were not observed at 17 days post-injection (Figure S1A). In addition, the expression of the 20S proteasome subunit was analyzed and no difference was found between both groups (Figure S1B). Finally,  the activity of the proteasomal catalytic unit 20Swas evaluated in its most activated site of degradation (LLVY) and a slight, but non-significant decrease (13%) in the proteasome  activity was observed in the CT26 group when compared to the CTRL group (Figure S1C).”

302  Do you need the word “delta” there?  Doesn’t the delta symbol in the figure indicate “change?”

306 “Representative photos comparing….”

325 Replace “since” with “because”

329-330   I suggest:  “…intervention two groups were used:  one remained untrained (termed CTRL group) and the second group underwent the AET protocol (termed AET group).”

330-332:  I suggest:  “On day 30, the CT26 tumour cells were injected into a subset of each of the two groups; that is, the CTRL and AET groups were each split into another two groups:….”

334  “…maintained for an additional….”

337  “…presented with…”

338-341  On the other hand,  effectiveness of the AET protocol in preventing cachexia was clear because neither a decrease in body mass nor a decrease in TA muscle mass was observed in the CT26 + AET group when compared to the CTRL group (Figure 2B and C, respectively).

349  “…thus a grip strength test was performed. Mice from CT26 group presented with a clear….”

354   “…because the slower tumour progression was obvious in….”

356  Replace “since” with “because…”

366-369  “No change in the phosphorylation levels of the ribosomal S6 protein, a direct target of mTORC1, was observed when the CT26 group and the CTRL were compared (Figure 2G and I).

377  Replace “capable”  with “able”

380-396:  I suggest 

Figure 2. Aerobic exercise training delays cancer cachexia progression. A: Experimental design used to study the preventive effects of the AET in the CT26 cancer cachexia model. Mice were previously trained for 30 days, and then they were injected subcutaneously with 106 CT26 cells. Thereafter, the AET protocol was maintained for 13 more days. B: Body mass analysed in the end of the protocol. It was observed that the CT26-bearing mice presented with a decrease in the body mass and the AET was effective in preventing this loss. C: TA muscles mass measured in the end of the protocol. It is clear to note that AET was able to prevent cancer-related muscle wasting. D: Maximum speed achieved in the MRC test performed in the end of the protocol. The CT26 group had a worse performance and the AET was able to prevent its drops. E: Maximum force evaluated by the grip strength test at the end of the protocol. A decrease in the force production by the CT26 group, which was prevented by the AET protocol, was observed. F: Tumour volume evolution throughout  the experimental protocol. The AET was able to delay the tumour progression, and also contributed to the benefits observed in the muscle tissue. G: Representative Western Blots images with experimental groups identified at the top. CTRL: healthy untrained group; CT26: tumour-bearing untrained group analyzed 13 days after the injections; AET:  healthy group trained along 43 days; CT26 + AET: tumour-bearing mice group trained along 43 days, in which 30 days were before being injected with the CT26 cells. H, I and J: Protein densitometry analyses of components of Akt/mTORC1 signalling  (Akt, phospho-Aktser473, S6, phospho-S6ser240/244, 4E-BP1 and phospho-4E-BP1thr37/46). K: Densitometry analyses of eIF-2α and  phospho-eIF-2αser51. Data are presented as mean ± SD. * p<0.05 vs. CTRL. # p<0.05; ## p<0.01 vs. CT26. CTRL, n=5-6. CT26, n=7-8. AET, n=4. CT26 + AET, n=7-8.

398-399  You state:  “we used two subsets of mice, one non-injected with the tumour cells (termed CTRL), and a second one injected with the tumour cells (termed CT26 group).”  Were the mice in the CTRL group injected with the same vehicle in which the tumor cells were suspended?  This would be appropriate design and it would be good to explain this here.

403  You state:  “mice were randomly divided into six different groups.”  I believe you should further explain this.  From the description, I assume you did this:  “The mice from each of the two groups (CTRL and CT26) were randomly assigned to one of six groups so that each of the six groups consisted of an even number of both CTRL and CT26 mice.”

413-414  fix awkward wording as suggested earlier.

Author Response

please see attachment the reply

Reviewer #3
Comments: The paper titled Aerobic exercise training and in vivo Akt activation counteract cancer cachexia by inducing a hypertrophic profile through eIF-3 2α modulation describes a set of very nice and inter-related studies, which investigate an extremely important topic - cachexia and the mechanism behind skeletal muscle loss. The paper highlights the important role of the AKT pathway in prevention of muscle loss and suggests aspect of this pathway could be therapeutic targets for muscle atrophy. This is extremely important because currently there are no really effective treatments for skeletal muscle loss. The experiments are well designed and nicely illustrated in the figures. The results are well-presented, thoroughly explained and are supported by the data in the figures. The methods are appropriate - although I have a question about the electro-transfer (see below). Over all this is an extremely nice paper, but the English needs improvement. I made some suggestions early in the paper (see below) and these changes in the sentence structures could be applied throughout the paper.

I do have a question about the Electro-transfer: GFP was used to ensure transfection occurred – I assume fluorescence was noted. However, was the group able to quantify this? It is not discussed. I ask because I wonder: How were corrections for differences in electro-transfer efficiency made? Obviously, one cannot assume that every muscle will “take up” and express the plasmids to the same extent. Often people will also co-electrotransfer a reporter gene (e.g., luciferase) that is quantifiable and compare this reporter gene expression between muscles as a measure of electrotransfer efficiency. It is clear in figure 3 that the AKT was expressed because there is a measurable effect that can only be attributed to that plasmid product. The data show that the expression was likely quite even from animal to animal. However, in figure S3, it is possible that the negative effect is a result of poor (or uneven) uptake and/or expression of the S6K1 plasmid. It is my understanding that the GFP was encoded by a separate plasmid that was co-electrotransferred with the plasmid encoding the S6K1. Would the authors comment on this, please? Perhaps I am missing something - that is possible. If so, I apologize.

Answer: The GFP plasmid was indeed used to ensure that the transfection had occurred. In general words, the green fluorescence was our guidance to check if the myofibers were or not positive for Akt or S6K1 in our experiments. This approach is widely used in in vivo electrotransfer experiments (Marabita et al; 2016, Gonçalves et al; 2019, Moretti et al; 2016) when the plasmids used are not carrying a luciferase reporter, which then would be quantified.
In our experiments GFP was encoded by a separate plasmid (pcDNA), which is an empty vector just to carry it into the myofibers together with the plasmid of interest, i.e., Akt or S6K1, however, the intensity of the green fluorescence is dependent of the capacity each myofiber has to express the new protein. In general words, some myofibers can appear with a strong green fluorescence while others can be found with a smoother green fluorescence, which is correct since the same can happen when a luciferase reporter is used.
The Figure 3 clearly shows that some myofibers could expressed Akt and then a huge increase in cross-sectional area (CSA) was observed. The Figure S3 shows that S6K1 did not induce an increase in the CSA. According to our understanding it happened because Akt is the major conductor of the pathway, i.e., once activated Akt can stimulated down-stream elements of the pathway, such as S6 and eukaryotic initiation factors, which in fact was proved and it is shown in the Figure 4. On the other hand, when we electroporated S6K1 plasmid in our cachectic mice it was not observed any changes in the CSA quantification. It was supposed happening because S6K1 is a downstream target of Akt/mTOR pathway, then to be stimulated it needs an upstream stimulus. A possible alternative could be an approach in which we could wait more time for the S6K1 plasmid been expressed into the myofibers, however, it was not possible since the CT26 cancer cells induce a fast cachexia.

Also, Figure 1 (line 304): That looks more like a linear rise than an exponential rise to me…. Both axes appear to be linear. Do you mean “rise steeply?”

Answer: Thank you for this observation. It was indeed a way to mean “rise steeply”. It is now corrected in the Figure legend 1 and can be found highlighted in yellow in this new version of the manuscript.

English Suggestions (by line number):

Legends should legend: “…each figure legend.” Answer: We have corrected it, please check line 96 in this new version of the manuscript (highlighted in yellow). Would read better as “At the end of each experimental procedure, ….” Rather than “In the end…”

Answer: We have corrected it, please check

lines 107 in this new version of the manuscript (highlighted in yellow). “…in order to verify the….”

Answer: We have corrected it, please check line 118 in this new version of the manuscript (highlighted in yellow).

“…and consisted of 60-minute….”

Answer: We have corrected it, please check line 123 in this new version of the manuscript (highlighted in yellow).

“A single exercise session to evaluate molecular changes induced by cancer cachexia was also performed.”

Answer: We have corrected it, please check lines 130-131 in this new version of the manuscript (highlighted in yellow).

“…experiments, a plasmid encoding Green fluorescent Protein (Snap-GFP) was co-transfected into the muscle.”

Answer: We have corrected it, please check lines 145-146 in this new version of the manuscript (highlighted in yellow). “…to overall threshold and brightness adjustments” - which were applied identically to all images?

Answer: The adjustments were applied identically to all of our images. We have corrected it, please check line 165 in this new version of the manuscript (highlighted in yellow).

“…composed of….”

Answer: We have corrected it, please check lines 181 and 186 in this new version of the manuscript (highlighted in yellow).

How were the hind limbs “fixed?”

Answer: The hind limbs were fixed by a ‘stabilization tower’ which tight fix the knee to the platform to avoid any kind of movement which could interfere in the muscle contraction. A simple representative scheme is showed below:

Would it be better to say “The data are presented as the mean +/- the standard deviation. Analyses of studies consisting of two groups were analyzed using….”

Answer: We have corrected it, please check lines 250-251 in this new version of the manuscript (highlighted in yellow).

I suggest: “A progressive decrease in the body mass of the tumour-bearing mice (CT26 group) was observed starting at 9 days after the cell injections and becoming significant on days 11 (11%), 13 (17%), 15 (20%) and 17 (21%) when compared to the healthy control non-injected mice (CTRL group) at the same time points (Figure 1B).”

Answer: We have corrected it, please check lines 262-265 in this new version of the manuscript (highlighted in yellow).

Is it a truly exponential rise?

Answer: Thank you once again for this observation. As mentioned above, it was a way to mean “rise steeply”. We have corrected it, please check line 271 in this new version of the manuscript (highlighted in yellow).

I suggest: “Because the main aim of this study was to better understand the skeletal muscle plasticity during cancer cachexia development, CT26-bearing mice were followed up until they presented with a significant body mass loss without signs of ulcerations in the tumour mass, thus avoiding any kind of tumour burden-associated suffering.”

Answer: We have corrected it, please check lines 273-276 in this new version of the manuscript (highlighted in yellow).

I suggest: “It is noteworthy that at 17 days post-injection no mice had died.”

Answer: We have corrected it, please check lines 279-280 in this new version of the manuscript (highlighted in yellow).

I suggest: “Thereafter, the fat mass and muscle mass content of both groups were assessed.” I suggest a similar sentence structure rearrangement as above.

Answer: We have corrected it, please check lines 280-284 in this new version of the manuscript (highlighted in yellow).

“….however, differences between both CTRL and CT26 groups were not observed at 17 days post-injection (Figure S1A). In addition, the expression of the 20S proteasome subunit was analyzed and no difference was found between both groups (Figure S1B). Finally, the activity of the proteasomal catalytic unit 20Swas evaluated in its most activated site of degradation (LLVY) and a slight, but non-significant decrease (13%) in the proteasome activity was observed in the CT26 group when compared to the CTRL group (Figure S1C).”

Answer: We have corrected it, please check lines 295-302 in this new version of the manuscript (highlighted in yellow).

Do you need the word “delta” there? Doesn’t the delta symbol in the figure indicate “change?”

Answer: We have corrected it, please check line 307 in this new version of the manuscript (highlighted in yellow).

“Representative photos comparing….”

Answer: We have corrected it, please check lines 311 in this new version of the manuscript (highlighted in yellow).

Replace “since” with “because”

Answer: We have corrected it, please check line 330 in this new version of the manuscript (highlighted in yellow).

I suggest: “…intervention two groups were used: one remained untrained (termed CTRL group) and the second group underwent the AET protocol (termed AET group).” I suggest: “On day 30, the CT26 tumour cells were injected into a subset of each of the two groups; that is, the CTRL and AET groups were each split into another two groups:….” “…maintained for an additional….” “…presented with…”

On the other hand, effectiveness of the AET protocol in preventing cachexia was clear because neither a decrease in body mass nor a decrease in TA muscle mass was observed in the CT26 + AET group when compared to the CTRL group (Figure 2B and C, respectively).

“…thus a grip strength test was performed. Mice from CT26 group presented with a clear….” “…because the slower tumour progression was obvious in….” Replace “since” with “because…” “No change in the phosphorylation levels of the ribosomal S6 protein, a direct target of mTORC1, was observed when the CT26 group and the CTRL were compared (Figure 2G and I).

Replace “capable” with “able”

Answer: Because their proximity throughout the text we have condensed all those suggestions and we have corrected all of that, please check lines 333-383 in this new version of the manuscript (highlighted in yellow).

I suggest

Figure 2. Aerobic exercise training delays cancer cachexia progression. A: Experimental design used to study the preventive effects of the AET in the CT26 cancer cachexia model. Mice were previously trained for 30 days, and then they were injected subcutaneously with 106 CT26 cells. Thereafter, the AET protocol was maintained for 13 more days. B: Body mass analysed in the end of the protocol. It was observed that the CT26-bearing mice presented with a decrease in the body mass and the AET was effective in preventing this loss. C: TA muscles mass measured in the end of the protocol. It is clear to note that AET was able to prevent cancer-related muscle wasting. D: Maximum speed achieved in the MRC test performed in the end of the protocol. The CT26 group had a worse performance and the AET was able to prevent its drops. E: Maximum force evaluated by the grip strength test at the end of the protocol. A decrease in the force production by the CT26 group, which was prevented by the AET protocol, was observed. F: Tumour volume evolution throughout the experimental protocol. The AET was able to delay the tumour progression, and also contributed to the benefits observed in the muscle tissue. G: Representative Western Blots images with experimental groups identified at the top. CTRL: healthy untrained group; CT26: tumour-bearing untrained group analyzed 13 days after the injections; AET: healthy group trained along 43 days; CT26 + AET: tumour-bearing mice group trained along 43 days, in which 30 days were before being injected with the CT26 cells. H, I and J: Protein densitometry analyses of components of Akt/mTORC1 signalling (Akt, phospho-Aktser473, S6, phospho-S6ser240/244, 4E-BP1 and phospho-4E-BP1thr37/46). K: Densitometry analyses of eIF-2α and phospho-eIF-2αser51. Data are presented as mean ± SD. * p<0.05 vs. CTRL. # p<0.05; ## p<0.01 vs. CT26. CTRL, n=5-6. CT26, n=7-8. AET, n=4. CT26 + AET, n=7-8.

Answer: We have corrected it, please check lines 385-401 in this new version of the manuscript (highlighted in yellow).

You state: “we used two subsets of mice, one non-injected with the tumour cells (termed CTRL), and a second one injected with the tumour cells (termed CT26 group).” Were the mice in the CTRL group injected with the same vehicle in which the tumor cells were suspended? This would be appropriate design and it would be good to explain this here.

Answer: Thank you for this observation. The CTRL group was indeed injected with the same volume (100ul) of serum free media. We have added it on Methods section, please check line 104-105 in this new version of the manuscript (highlighted in yellow).

You state: “mice were randomly divided into six different groups.” I believe you should further explain this. From the description, I assume you did this: “The mice from each of the two groups (CTRL and CT26) were randomly assigned to one of six groups so that each of the six groups consisted of an even number of both CTRL and CT26 mice.”

Answer: Thank you for this observation. That is exactly the meaning of our sentence, mice from each group (CT26 and CTRL) were randomly assigned in one of those six subgroups, so each subgroup had the same number of mice.

fix awkward wording as suggested earlier.

Answer: We have corrected it, please check lines 417-422 in this new version of the manuscript (highlighted in yellow).

Finally, we would like to thank you for your help in evaluating this manuscript and trust that this new revised version will be acceptable for publication in the CANCERS.
